# Testing of Eluates from Waterproof Building Materials for Potential Environmental Effects Due to the Behavior of *Enchytraeus albidus*

**DOI:** 10.3390/ma14020294

**Published:** 2021-01-08

**Authors:** Marya Anne von Wolff, Stephan Pflugmacher, Dietmar Stephan

**Affiliations:** 1Group of Building Materials and Construction Chemistry, Department of Civil Engineering, Technische Universität Berlin, Gustav-Meyer-Allee 25, 13B, 13555 Berlin, Germany; vonwolff@tu-berlin.de; 2Joint Laboratory of Applied Ecotoxicology, Environmental Safety Group, Korea Institute of Science and Technology Europe (KIST Europe), Stuhlsatzenhausweg 97, 66123 Saarbrücken, Germany; stephan.pflugmacher@helsinki.fi

**Keywords:** *Enchytraeids*, waterproof building materials, ecotoxicology, biotest

## Abstract

In order to determine the potential environmental impact of construction products, it is necessary to evaluate their influence on organisms exposed to them or their eluates under environmental conditions. The behavior of the white worm *Enchytraeus albidus* is a useful tool for assessing the potential environmental impact of construction products in contact with water and soil. This study investigates the environmental effects of eluates from two construction products, a reactive waterproofing product, and an injection resin, on the reproduction and avoidance behavior of *E. albidus*. The eluates were prepared according to existing guidelines. The soil used for the tests was moistened with the eluates of the construction products. The reproduction results of the worms were collected after six weeks of exposure. Offsprings were counted under the microscope and statistically analyzed. Results from the avoidance behavior were collected after 48 h of exposure, and results were compared with the reproduction results. The eluates from both construction products induced significant changes in the reproduction behavior of *E. albidus*. Undiluted or only slightly diluted eluates of the injection resin drastically reduced the reproduction of the worms, whereas the leaches of the reactive waterproofing product only had a minor effect. The avoidance results for the injection resin indicates that its presence in the habitat is clearly detrimental to the survival of *E. albidus,* while the avoidance results for the waterproofing resin showed an initial avoidance of the eluates, but no harmful effects were observed. The avoidance test is a way of rapid toxicity screening of environmental samples when time is a critical parameter to measure possible environmental effects. This study shows that ecotoxicological tests using *Enchytraeids* are a valuable and important tool for understanding the mode of action of eluates from construction products in the environment.

## 1. Introduction

The environmental impacts of existing construction materials are important to consider while developing new products. Once the construction products are exposed to weathering, they could potentially be leached, and the resulting eluates could have a negative impact on organisms in the environment [1]. The chemical composition of construction materials and their leaching behavior are crucial for the environmental compatibility of a product [2].

In addition to this prospect, other stages of the building material also play a role in its environmental compatibility. Even during the construction phase, dismantling, recycling, and disposal [3,4], water-soluble substances can be released, especially from fine-grained materials, which have an impact on the environment [5]. As all building materials have a limited life span and weather over time, the assessment of leaching behavior and the evaluation of the environmental relevance of the resulting eluates are of great importance for the certification of building materials in the European market [6].

Dynamic surface leaching test (DSLT), according to CEN/TS 16637-2:2016 [7], is widely applied in Europe to determine the leaching behavior of construction materials and is recommended by the German Environmental Agency (Umweltbundesamt, UBA- Berlin, Germany) for their environmental impact assessment [8]. The DSLT is one of the standard methods to evaluate the release of dangerous substances from building products. The test determines the release of inorganic and non-volatile organic substances through contact with leaching agents per unit area of the construction product under investigation as a function of time. During the leaching process, 8 eluates are generated under specified test conditions, which are then examined for chemical and ecotoxicological parameters. The end of the standard experiment is the 64th day of the experiment, and the release of substances related to the specific surface is determined.

Several studies have investigated the release behavior of inorganic ions such as Na^+^, Al^3+^, Ca^2+^, Si^4+^, and Cu^2+^ and organic substances (e.g., superplasticizers) from concrete materials and construction waste [8,9,10,11]. However, only a few are taking the results of an ecotoxicological analysis into account [12,13,14].

Currently, the most used toxicological bioassays for testing products are made using species as *Daphnia sp*., Algae, and *Danio rerio*. Although those species are very representative for the aquatic environments, the scenarios of construction products are often in the terrestrial environment, turning those species not always the ideal match for the necessary bioassays. Ecological risk assessment from construction materials can be examined by aquatic and terrestrial biomarkers [15], among others, the ecotoxicological tests using the Oligochaeta species of genus *Enchytraeidae*. Although ecotoxicological tests using these species have been developed and standardized only within the last two decades, it became an important indicator organism for the determination of impacts on soil ecosystem due to their sensitivity to a broad spectrum of xenobiotics, ease of maintenance in the laboratory [15], and widely representation worldwide [16,17,18].

Between the possible ecotoxicological tests available, two are specifically used: the reproduction and the avoidance behaviors. The reproduction test using Oligochaetas is very commonly used, but it takes much more work and is very time-consuming since the complete ecotoxicological test takes up to 6 weeks of work and constant measurements. The avoidance behavior test is much simpler compared to the reproduction test and is also less time-consuming since an avoidance test takes only 48 h to get results. As shown in previous studies, avoidance behavior can be used as a first indication of the occurrence of damage in *Enchytraeids* when exposed to nanoparticles [19], giving a very fast result when compared to the traditional studies using reproduction tests.

However, although faster results can be obtained using test methods as avoidance behavior, the reproduction tests with *Enchytraeids* seem to be the most accurate in looking at the long term exposure scenario [20] and is likewise the best way to associate with real field exposures of chemicals from anthropogenic activities [21].

Tests performed in this study intended to fill the information gap about exposures of Oligochaetes in the presence of eluates from construction product particles. This study aims to verify the possible effects of waterproof building materials in the environment using the reproduction and avoidance behavior as ecotoxicological tools [22] to elucidate the effects of leaches from construction materials prepared with different types of water for the environment.

## 2. Materials and Methods

### 2.1. Construction Products

This research selected two examples of construction products that are applied directly in contact with soil or water.

The first construction product used was a waterproof material to protect walls from water ingress, for example. The non-commercial material consisted of 2 components, mixed by hand with a spatula in the mass proportion of 1:1 in a beaker. The resulting paste was placed in silicone forms and cured for 46 days until the product was completely dry.

The second construction product is a very fast-reacting silicate injection resin. The injection resin also consists of 2 components, which have to be mixed within a few seconds. These were filled in two-chamber cartridges. A static mixer was screwed to the outlet of the cartridges, and the cartridge was placed on a compressed air gun. The components react immediately, and the mixture was extruded with the air-gun in silicone molds, where it hardens within a few minutes. Although the material cures very quickly due to the reaction heat, in this study, we waited for 7 days for complete stabilization of the product after the reaction before submerging in the water baths for leaching. Due to a long time for leaching and constantly washing the pieces on the baths, all the eluates were tested without further dilution.

Since the two products cover very different areas in terms of their structural application, the results are not intended to be compared either. At this point, it must be pointed out again that special laboratory formulations with known ingredients were deliberately worked on and not on ecotoxicologically tested and marketable products, so that effects can be achieved in the comparative investigations with a high probability.

### 2.2. Leaching Method

Leaching of the two construction materials was performed according to CEN/TS 16637-2 [7]. The decision to prepare the eluates with two different types of water was taken to search for the best conditions for survival and thus for the reproduction of the *Enchytraeids*. The eluates were prepared using tap water and distilled water as the medium for leaching the components from the construction products. The pieces of the construction products previously prepared according to item 2.1 were submerged in baths of distilled water or tap water. The dimensions of the container used for leaching was 31 × 23 × 9 cm^3^ (l × w × h) and has a capacity of 3.5 L. Eluates from both construction products were collected after exposure times of 6 h and 1, 2, 4, 9, 16, 36, and 64 days, and vacuum filtered using a micropore filter of 0.45 µm. All eluates were characterized for pH, electric conductivity, and the following inorganic components: Al^3+^; Ca^2+^; Cu^2+^; K^+^; Na^+^; Si^4*^; NO_3_^−^; SO_4_^2−^; Cl^−^. The leaching procedure was performed in triplicates, and results are presented further with the mean values obtained for the parameters.

### 2.3. Test Organisms

This study used the species *Enchytraeus albidus* as the test organism [22]. Worms were cultured for many years in the ecotoxicological laboratory of the Technische Universität Berlin using a bio garden soil, kept at a controlled temperature of 10 °C and fed at libitum with bio rolled oats, autoclaved and finely grounded. Organisms were cleaned from soil particles and acclimatized before starting the test procedure.

### 2.4. Soil

Standard soil (LUFA 2.2) was used to perform the avoidance and reproduction tests. The soil was commercially acquired from Landwirtschaftliche Untersuchungsund Forschungsanstalt (LUFA) Speyer, Germany [23]. The characteristics of the soil are: soil type: loamy sand; dry matter of the soil: 94.8 wt.%; water content 5.4 g water/100 g soil; maximum water holding capacity (WHC): 44.8 ± 2.9 wt.%; pH: 5.6 ± 0.4; cation exchange capacity: 9.2 cmol/kg ± 1.4.

### 2.5. Experimental Design

#### 2.5.1. Reproduction Test

Rounded glass vessels with 100 mL were used for all the reproduction tests. In each vessel, 50 g of soil was placed inside and moistened using the eluate until the maximum water holding capacity of 46 wt.%. As control sets, the same conditions were placed using three different types of water: tap water, distilled water, and reconstituted freshwater. All the control vessels were completely free of leaching from the construction products. For each condition tested, four replicates were placed [21].

Ten adult *Enchytraeids* individuals with a well-developed clitellum were placed in each vessel. All vessels were covered with a lid containing small holes in order to avoid escaping the worms. The worms in each vessel were fed with 0.2 mg of bio rolled oats per week, distributed equally. Vessels were weighed and kept at a controlled temperature of 20 °C, and water content was replaced when evaporation occurred. The experimental plan ran for six weeks. After the first four weeks of exposure, the adult *Enchytraeids* were removed, and vessels were kept at the same conditions in order to wait for the hatching of the cocoons. After additional two weeks, organisms were fixed and colored using the extraction method of staining with Bengal red according to ISO 16387:2014 [21] and counted under the microscope. The results obtained from the tests were compared to the control results.

#### 2.5.2. Avoidance Test

To perform the avoidance tests, once again, rounded glass vessels with 100 mL were used. The tests were performed according to ISO 17512 [19]. The vessels were divided into two sections using a removable wall. On one side, was placed 25 g of LUFA soil 2.2, moistened until the maximum water holding capacity of the soil was reached using the eluate of the construction product. On the opposite side of the vessel, LUFA soil moistened with the same corresponding type of water was placed. That means, if the eluate from the construction product was previously prepared with tap water, the soil on the control side was also moistened with tap water, but without any previous contact with the construction product. This was an attempt to prove the avoidance behavior by the possible components leached and not just by the type of water.

After placing both soils on the vessel, the wall was removed, and 10 *Enchytraieds* per vessel with a well-developed clitellum were introduced in the fine line that divides both soils. The vessels were covered with a lid containing small holes to permit air exchange. Four replicates per treatment were prepared, and vessels were left at a temperature of 20 °C and period light control (16/8—light/dark) for 48 h without food.

After 48 h, a removable wall was again introduced in the division of the soils and both sides were searched individually for the worms. Worms were counted, and results were compared. Figure 1 shows the schematic representation of the avoidance test, with all steps.

Negative control was also placed where both sides of the vessel contained the same type of control to evidence the non-avoidance behavior when both sides contain the same component.

## 3. Results and Discussion

### 3.1. Eluate Characteristics

The values measured for pH and electric conductivity of the eluates are represented in Figure 2. The chart represents the mean values obtained from the triplicates made for each day of measurement for the following eluates: Silicate resin product distilled water (DW) and silicate resin product tap water (TW) and also for the waterproof product distilled water (DW) and waterproof product tap water (TW).

The variation of electric conductivity of the silicate injection resin for DW presented a variation between 1.5 and 3.5 mS/cm. This variation can be explained due to the chemical behavior of the silicate building material when exposed to constant contact with water [24]. The range variation of the electric conductivity of eluates from the waterproof material with TW and DW was very stable, reaching a maximum value of 1 mS/cm during the total time of leaching.

The eluates with TW and DW of the silicate injection resin showed very similar behavior in the variation of the pH, starting with values of pH 10 and slowly stabilizing until pH 8. The pH of the eluates from the waterproof material in TW and DW presented a small variation of results along the 64 days of leaching alternating between pH 8 and 9.

Table 1 presents the mean values of inorganic content analyzed for the eluates of the construction products and the blank samples. All samples were analyzed in triplicates by inductively coupled plasma atomic emission spectroscopy ICP-AES method.

The highest variations in analogy with the correspondent blank sample occurred for the silicate product TW, indicating that the constant baths of the pieces stimulated the increase of components as Al^3+^; Ca^2+^, Na^+^, Si^4+^, and NO_3_^−^. The same occurred for the silicate samples leached with DW; however, the increase was proportional to the initial presence of inorganic components. For the waterproof product in TW and DW, inorganic components as Ca^2+^ and Na^+^ were leached out.

The DSLT has already been used in many studies. Brameshuber et al. [25] applied this to the mortar to investigate the release of organic constituents from concrete under practice-relevant conditions, whereby the effects of organic substances in the eluates could be classified as minor. However, contamination with sodium, sulfates, aluminum, and some heavy metals could be proven while using contaminated concrete blocks. As part of the selection of the processes suitable for the ecotoxicological assessment of building materials [13], the DSLT was carried out on 37 representative building products that contained mobilizable organic substances. The point of criticism of the DSLT was that some substances could no longer be identified due to their biodegradability over the duration of the DSLT of 64 d. This argument is also put forward by Bandow et al. [1] because the conversion of organic substances cannot be excluded, especially under real conditions. The transferability of the test results from the DSLT to real environmental and practical conditions also appears problematic. However, in its principles for evaluating the effects of construction products on soil and groundwater [26], the Deutsche Institut für Bautechnik already states how the laboratory results from the horizontal leaching test can be transferred to real conditions through model considerations. Scherer described the DSLT as “the authoritative and recognized test procedure for evaluating the environmental impact” [27]. Due to the frequent use of tests, test variants modified by the DIN standard are already available. Märkl et al. [12] adapted the DSLT for plastic products and used it to leach polyurethane resin during the curing phase. Within the research project, the DSLT was used both for reactive sealing and for a silicate injection resin. As described in DIN CEN/TS16637-2, the DSLT applies to evaluating the surface-dependent release for monolithic, plate-like, and film-like products. The type of construction product determines the implementation conditions and the dimensions of the test specimen. The waterproof product was classified as a plate-like product, while the silicate injection resin was classified as a monolithic product.

### 3.2. Reproduction Results

In order to determine the best control design for the reproduction behavior and the subsequent comparison of the results with leachings of the construction products, this study collected data of three different scenarios of control sets using tap water, distilled water, and reconstituted freshwater as watering medium of the LUFA soil. The results presented in Figure 3 evidence the highest number of offsprings for the vessels moistened with reconstituted freshwater, while a very similar result was achieved using tap water. Based on these results, which were combined to simulate real construction scenario situations, this study carried out the leaching of the construction components with tap water and distilled water.

After the total exposure time of 6 weeks, the offsprings of *E. albidus* were counted, and the results are shown in Figure 4 and Figure 5. The control represents the mean number of *Enchytraeids* on the water reproduction test, where no eluates of construction products were in contact with this group.

An analysis of Figure 4a shows a low decrease in *Enchytraeids* reproduction behavior. Nevertheless, after 16 days of leaching, an apparent decrease in the number of offsprings can be seen, with less than 15 worms counted. The same decrease was perceived for the eluates of the waterproof material with DW after 16 days, (Figure 4b), however with a total amount of 17 *Enchytraeids*. The most significant results are emphasized with an (*) point.

The reproduction tests were performed for all the eluates. When observing the reproduction results for both eluates, after 64 days, a stabilization of the reproduction behavior was noticed at the end of the test, indicating a reduction of the toxicity and decrease in the presence of active ingredients from the waterproof material when exposed to water for a longer time [28]. The drop in the number of offsprings was not very alarming for eluates of the waterproof material. However, effects on the *Enchytraeid* populations can be noticed. An alarming situation would be configurated when the component tested affects the reproduction behavior in an exposed concentration that affects fifty percent of the population. The different results over time represent typical leaching behavior. In the beginning, there was a first wash-off followed by reduced diffusion-controlled release.

Both types of eluates (TW and DW) of the waterproof material contained a low variation in the quantities of ions when compared to the corresponding blank samples. Ions of Na^+^ and Ca^2+^ increased with the leaching. However, it is already proven that sodium and calcium have a small influence on toxicological effects for *Enchytraeids* [29], and the presence of these minerals on the leachings can help to clarify the reduction of offsprings in the results.

Analyzing the results of the eluates of the silicate resin are plotted in Figure 5a,b. A high impact of the eluates from the silicate product on the reproduction behavior of the *Enchytraeids* can be seen. In all phases of the leaching process, small numbers of offsprings were counted. The silicate eluates prepared with distilled water presented the highest impact on the population, bringing special attention to the reproduction test from day 9 silicate DW, where a mean number close to 0 was achieved. For both reproduction tests of silicate leachings from day 64, the *Enchytraeids* presented a tendency to recover the population of worms, and a higher number of offsprings was counted; however, the number of offsprings is still under fifty percent of the population when compared to the control group. The most significant results are emphasized with an (*) point.

The silicate leaching for DW on day 16 presented a slight recovery of the results compared to the previous reading on the same test. This effect can sometimes occur once the *Enchytraeids* are biological indicators, and the sensibility can change according to the exposure [30]. Few other details were also visible during the reproduction test days, e.g., some of the silicate eluates revealed the presence of a few eggs, but these were too weak to hatch and did not hatch until the end of the test. In other situations, the initially introduced adult worms used died at the beginning of the test or even one week before the end of the test. Besides, the absence of adult worms had a direct effect on the number of offsprings.

Besides the increased concentration of sodium, chlorides, and sulfates in the eluates from the silicate component, it is impossible to confirm with certainty what causes the strong toxicity influence from the eluates from the silicate resin for the *Enchytraeids*. The very small number of offsprings, in one factor, can be explained by the sensitive sensors on the body of the *Enchytraeids*, which perceive limited survival conditions early on and will avoid habitats [31].

### 3.3. Avoidance Results

The same eluates were tested for the avoidance behavior of the *Enchytraeids,* and results are represented in the following graphics. Figure 6 represents the avoidance behavior of the *Enchytraeids* using the same treatment on both sides (negative control).

Figure 6 shows the graphic proportion of worms that prefer to stay on each side of the soil. On both sides, the same control conditions were used, and it was possible to see that when both sides contain a control soil free of toxicants, the worms do not show one preferred side. The same average distribution was observed for both sides, being one side with 49% of the population and the other side with 51% of the population of *Enchytraeids*. According to the ISO 17512 (2008), the control is considered valid if the proportions stay between 40–60% in the distribution hack.

The avoidance results for the waterproof leachings are plotted in Figure 7a,b. The green line represents the number of *Enchytraeids* that chose the control side, while the red line represents the number of *Enchytraeids* that preferred the eluate side.

Analyzing the avoidance behavior of the *Enchytraeids* testing the waterproof leaches, it was possible to see that the worms tend to prefer mostly the side of the control, but for few leaches, the difference in the number of worms in each side is small. A less poor scenario in the number of organisms on each side is achieved in the vessels of waterproof TW 64 days. The statistical avoidance percentage was calculated as represented in Table 2.

Figure 8a,b represents the avoidance behavior of the *Enchytraeids* for the eluates of the silicate. The result indicates that one more time, the *Enchytraeids* tend to move mostly to the side of the control soil, where no contaminations of chemicals are detected. The avoidance was similar in both tests, with few variations in numbers for 9 and 64 days.

A statistical analysis was carried out to calculate the percentage of worms affected. Avoidance was calculated according to the following Equation (1):(1)$$x=\left(\frac{n_{c}-n_{t}}{N}\right)\times 100$$
where:*x* is avoidance, expressed as a percentage;*nc* is the number of worms in the control soil (either per vessel or in the control soil of all replicates);*nt* is the number of worms in the test soil (either per vessel or in the test soil of all replicates);*N* is the total number of worms (usually 10; either per vessel or in the control soil of all replicates).

Using the results collected on the avoidance tests, the percentage of avoidance was calculated and the results are represented in Table 2.

According to ISO 17512, when the avoidance results are ≥80%, a limited survival habitat function is configurated. If an attraction of >80% by the test soil is observed, the presence of chemical substances cannot be excluded. The result indicates an impact on the behavior of the organisms.

The limited survival habitat was determined on the following eluates: silicate TW (6 h, 1 day, 9 days) and silicate DW (6 h, 1 day, 9 days). Although eluates from waterproof did not represent a limited habitat function, in few eluates, the avoidance percentage reaches 75% being very close to this value and indicating the possible presence of chemicals.

The concentration of harmful substances consequently seems to be significantly increased in the first leaching steps since many components that can be leached out at the beginning of a DSLT can be released through washing. With the length of the leaching times, the number of Enchytraeids that migrate to the eluate side increases, indicating a decreased release of harmful substances with the time of leaching.

A better distribution of the number of organisms on both sides of the vessels for all leachings is noted in leaches of 64 days when an avoidance rate of 30% and 50% is reached.

### 3.4. Evaluation of Enchytraeid Results—Reproduction Versus Avoidance

Comparing the results of the reproduction tests with the results of the avoidance tests, a certain similarity can be observed in the sensitivity of both test results.

The initial eluates of the silicate product caused an impacting reduction on the number of *Enchytraeids* in the reproduction test, while in the avoidance test, the same eluates indicated an avoidance of 85% of the population.

Besides, the eluates from waterproof for reproduction tests did not present high toxicity for the organisms, but a high avoidance percentage was observed for the same components. It is important to remember that the reproduction test conditions supply constant contact with the soil + eluate for a chronic exposure time, allowing the leaches to stabilize for a longer period of time [32].

For all eluates of 64 days, in the reproduction test, an increase in the number of offsprings was observed. At the same time, the avoidance test also shows better survival habitat conditions with smaller avoidance behavior, being both test results an indicator of stabilization of chemicals in the soil.

The reproduction test is applied to detect effects resulting from sublethal concentrations and long term scenario exposures. The avoidance test exists to investigate the habitat function of soil with earthworms as representatives of the soil biocenosis. The endpoints are determined to obtain information on the environmental effects. The reproduction test is very labor-intensive and time-consuming, needing long incubation periods and results being collected after 6 weeks of exposure and constant work, turning the reproduction tests more expensive. In contrast, the avoidance test presents very fast results of environmental effects and a high level of sensitivity. However, the avoidance test is not intended to replace the reproduction test but to provide faster screenings of environmental effects in different levels of sensitivity. For a complete and better understanding of the effects of a toxicant on the soil, it is possible to use the avoidance test as the first information screen and the reproduction test to verify the chronic sublethal effects.

## 4. Conclusions

The reproduction behavior of *Enchytraeus albidus* presented the highest numbers when worms were in soil moistened with the reconstituted freshwater as a control, followed by tap water and, in last, the demineralized water. The type of water used to leach the components influenced the reproduction behavior of *E. albidus*. The best performance in the number of juveniles is achieved when the standard soil is moistened with liquids containing physic-chemical conditions more similar to those found in the environment.

Evaluating the reproduction results, it can be concluded that the eluates from the waterproof material did not present high toxicity to the reproduction behavior of *E. albidus;* besides, a decrease in the number of juveniles was measured. The leachings from waterproof TW presented a higher number of juveniles in comparison to waterproof DW. The leachings from silicate TW and silicate DW were toxic for the reproduction behavior of *E. albidus*. Eluates from silicate DW had the lowest number of juveniles and inhibited more than 50% of the population of *E. albidus*.

The avoidance results for all the eluates presented a similar sensitivity for *E. albidus* when compared to the reproduction results. The organisms avoided soils containing high concentrations of chemicals. Eluates of the silicate resin product in soil presented a limited habitat function with avoidance results >80%.

Looking at these results, it can be concluded that it is important to track biomarkers for *Enchytraeids* in order to assess the possible hazard effects of construction products and understand the mechanism of action in the environment.

## 5. Final Comments

The time and effort required for the preparation of the leachings in accordance with DIN CEN/TS 16637-2 should also be emphasized: it is extensive and intensive work, and therefore this study recommends for screening the use of other methods of leaching as the EN 12457-4 [33], which lead to similar results from a biological point of view.

It is important to note that a difference in the final products and consequently in the eluates may occur due to different handling of the starting products. The air gun and mixers used in the laboratories are prepared to handle a much smaller sample volume than those used in the construction industry. For this reason, it is recommended here to collect samples from real applications in order to clarify whether the products have possible toxicity. Another problem is the solubility of these components in the environment. The laboratory tests try to consider as far as possible the aspects from a real scenario situation, but it is still likely that this will not give the same results as in the real environment and that a much higher dilution of these components would occur and, therefore, lower toxicity would be detected.

The organisms used in these tests may never come into contact with this type of component in a real scenario. Nevertheless, it is the responsibility of ecotoxicology to identify the possible causes and influences that could occur in case of such an event. This study does not condemn the use of construction products, nor does it attempt to restrict or prevent their use. The sole purpose is to clarify human activities and their consequences for the environment and to find sustainable solutions that would help to focus on satisfying the needs of the present without compromising the ability of future generations to meet their needs.

## Figures and Tables

**Figure 1 materials-14-00294-f001:**
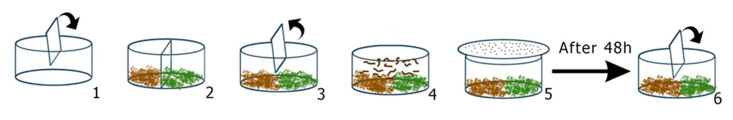
Schematic representation of the experimental procedure of the *Enchytraeid* avoidance test: (**1**) inserting the movable wall into the center of the test vessel; (**2**) introduction of the soils to be tested; (**3**) the movable wall is removed; (**4**) placing the *Enchytraeids* in the middle of the soil; (**5**) covering the test vessel with a lid (perforated); (**6**) reintroduce the wall to separate the floors and count the organisms present on each side.

**Figure 2 materials-14-00294-f002:**
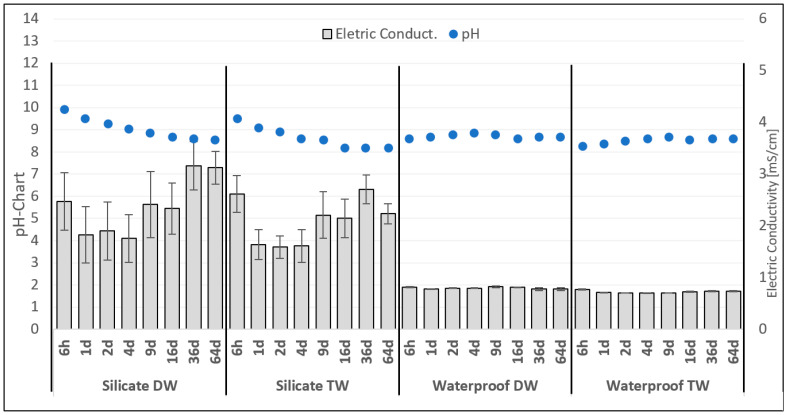
Graphic representation of pH and electric conductivity. Values are measured for all eluates of silicate and waterproof products.

**Figure 3 materials-14-00294-f003:**
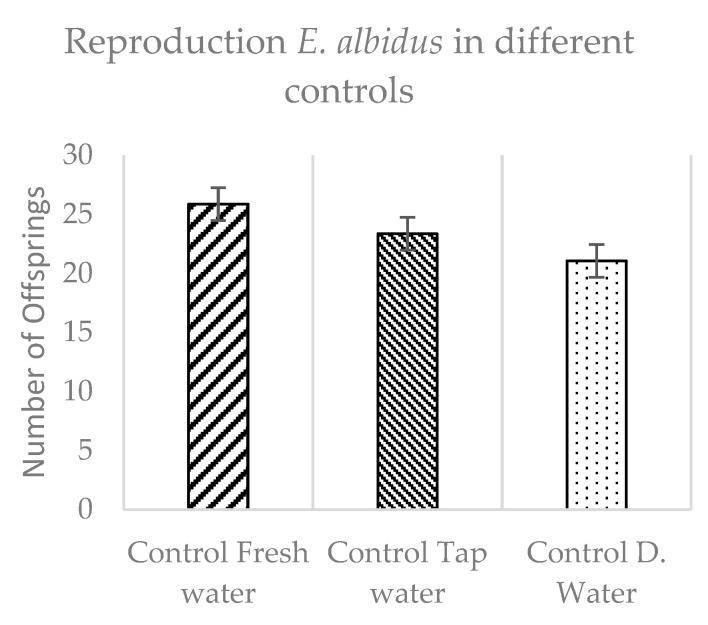
Reproduction behavior of *E. albidus* comparing three different types of water.

**Figure 4 materials-14-00294-f004:**
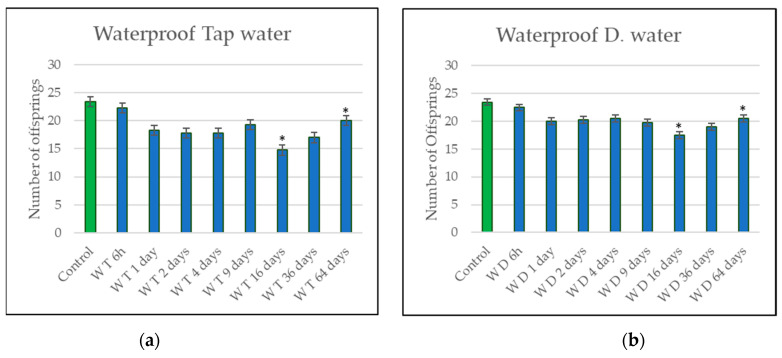
(**a**) *Enchytraeid* reproduction results for eluates from the waterproof material using tap water. (**b**) *Enchytraeid* reproduction results for eluates from the waterproof material using distilled water. Significant results are marked with an (*).

**Figure 5 materials-14-00294-f005:**
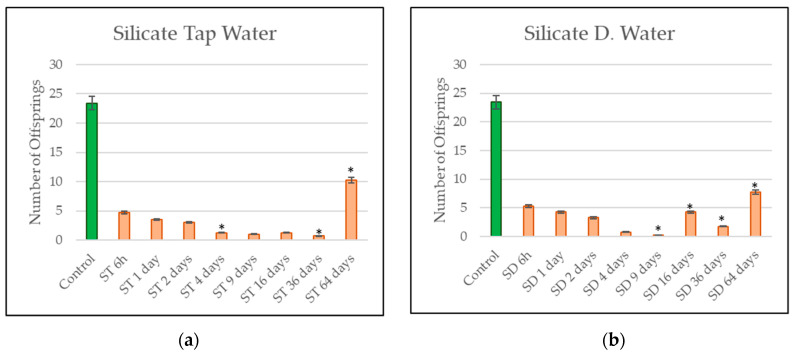
(**a**) *Enchytraeid* reproduction results for the silicate leach using tap water. (**b**) *Enchytraeid* reproduction results for the silicate leach using distilled water. Significant results are marked with an (*).

**Figure 6 materials-14-00294-f006:**
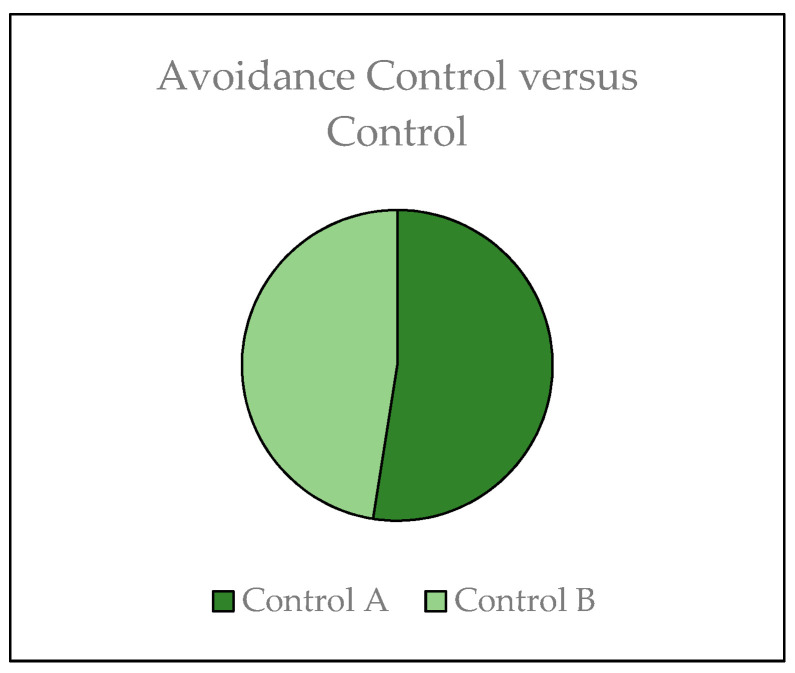
*Enchytraeid* avoidance distribution for both sides with control soil.

**Figure 7 materials-14-00294-f007:**
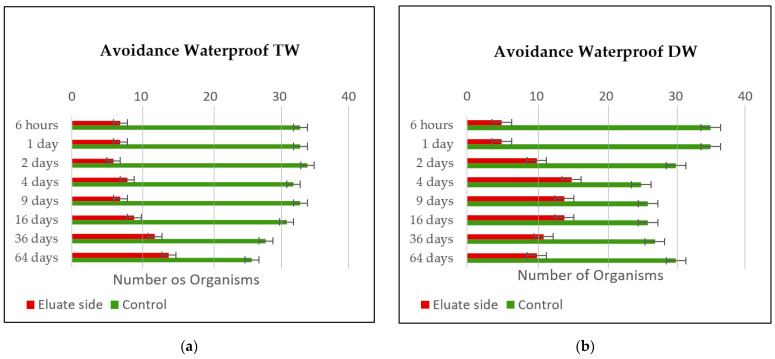
(**a**) *Enchytraeid* avoidance results for the waterproof leach using tap water. (**b**) *Enchytraeid* avoidance results for the waterproof leach using distilled water.

**Figure 8 materials-14-00294-f008:**
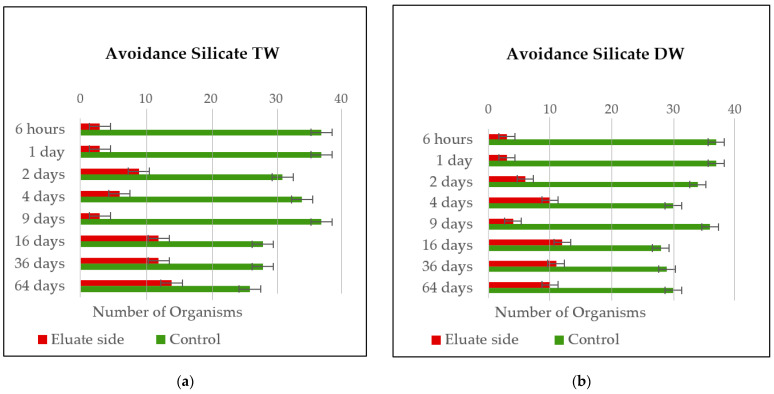
(**a**) *Enchytraeid* avoidance results for the silicate leach using tap water. (**b**) *Enchytraeid* avoidance results for the silicate leach using distilled water.

**Table 1 materials-14-00294-t001:** Inorganic characterization of the eluates: mean values for leachings and blanks in mg/L.

Parameter	Silicate TW	Silicate DW	Waterproof TW	Waterproof DW	Blank TW	Blank DW
Al^3+^	0.34	0.14	0.5	0.1	0.01	0.001
Ca^2+^	50.2	47.6	123.5	53.8	1.6	0.9
Cu^2+^	0.6	0.5	0.9	0.1	0.1	0.1
K^+^	11.0	10.1	32.7	7.9	1.1	0.2
Na^+^	1193.6	771.8	232.5	58.6	181.1	9.1
Si^4+^	53.8	48.6	8.2	6.8	6.2	0.2
NO_3_^−^	11.3	9.3	6.4	1.2	5.7	0.3
SO_4_^2−^	167.5	111.4	136.5	14.6	102.4	3.4
Cl^−^	66.8	57.8	64.8	53.7	49.8	2.2

**Table 2 materials-14-00294-t002:** Avoidance results percentages (%) for *E. albidus,* calculated for all eluates of waterproof and silicate.

Time	Waterproof TW	Waterproof DW	Silicate TW	Silicate DW
6 h	65	75	85	85
1 day	65	75	85	85
2 days	70	50	55	70
4 days	60	25	70	50
9 days	65	30	85	80
16 days	55	30	40	40
36 days	40	40	40	45
64 days	30	50	30	50

## Data Availability

Data availability within this article.

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
