# Peer review of "Testing of Eluates from Waterproof Building Materials for Potential Environmental Effects Due to the Behavior of Enchytraeus albidus"

_materials, 2021, doi:10.3390/ma14020294_

Round 1
Reviewer 1 Report
The paper is interesting; the approach is innovative, but I doubt it is within the scope of the Journal, as confirmed by the keywords. This is the reason I reject the paper.
The title is not compliant with the content: it refers to "constrution products" (the reader thinks a lot of products), while only two construction products, a reactive waterproofing product, and an injection resin, have been investigated as cause of modfication of reproduction and behavior of E. albidus.
In the introduction "phase" of construction products does not refer to the European standard EN15804 which defines the product category rules for construction products and the "phases" during their service life.
Only at lines 92 and 93 (in the Materials and Methods section) the reader knows that the research considers construction products applied directly in contact with soilor water.
Lines 136-139 could be tabled.
Abscissa in Figure 2 should be represented in scale: the bar graph is not suitable.
Line 352: the Equation should be numbered.
Author Response
Dear reviewers,
thank you for giving us the opportunity to submit a revised draft of our manuscript titled “Testing of eluates from construction products for potential environmental effects due to the behaviour of Enchytraeus albidus” for the Special Issue "Measurement of the Environmental Impact of Materials" within the Journal Materials – MDPI. We appreciate the time and effort that you have dedicated to providing your valuable feedback on our manuscript. We are grateful to the reviewers for their insightful comments on our manuscript. We have been able to incorporate changes to reflect most of the suggestions provided by the reviewers. In the following text, we have explained the changes within the manuscript according to the comments of the reviewers.
In the attached file is a point-by-point response to the reviewers’ comments and concerns.
We appreciate your collaboration and we are looking forward to hearing from you soon.
Sincerely,
MSc Marya Anne von Wolff
Prof. Dr. Stephan Pflugmacher Lima
Prof. Dr. Dietmar Stephan

Reviewer 2 Report
The subject of eluates from construction products for potential environmental effects is actual and important. It is particularly noteworthy that two fields of science have been merged in order to assess the phenomenon. This approach broadens research horizons.
The article is very valuable and prepared with great care, which can be seen especially in the detailed and very clear description of the graphs.
The introduction provide background and include relevant references, In a gentle manner, gradually introducing the reader to the complex subject.
The research is design and perform appropriate. All methods are described adequately. Results are present clearly, with the presentation. However, the study does not condemn the use of construction products, nor does it attempt to restrict or prevent their use. Article only clarify that human activities and their consequences for the environment and to find sustainable solutions that would help to focus on satisfying the needs of the present without compromising the ability of future generations to meet their needs. In my opinion, the scientific work should be extended to include an attempt to give the potential practical application of Enchytraeus albidus in engineering practice. Perhaps the authors could suggest some kind of study to assess construction products using the method presented.
To sum up, I think that the article deserves attention, because it is a very well done and presented scientific work. In my opinion the article can be published after including the practical application of the presented methodology.
Author Response

(The authors gave the same response as above.)

Round 2
Reviewer 1 Report
the paper can be accepted